# STOCHASTIC CONDITIONAL GENERATIVE NETWORKS WITH BASIS DECOMPOSITION

**Ze Wang, Xiuyuan Cheng, Guillermo Sapiro, Qiang Qiu**
Duke University
`{ze.w, xiuyuan.cheng, guillermo.sapiro, qiang.qiu}@duke.edu`

## ABSTRACT

While generative adversarial networks (GANs) have revolutionized machine learning, a number of open questions remain to fully understand them and exploit their power. One of these questions is how to efficiently achieve proper diversity and sampling of the multi-mode data space. To address this, we introduce BasisGAN, a stochastic conditional multi-mode image generator. By exploiting the observation that a convolutional filter can be well approximated as a linear combination of a small set of basis elements, we learn a plug-and-played basis generator to stochastically generate basis elements, with just a few hundred of parameters, to fully embed stochasticity into convolutional filters. By sampling basis elements instead of filters, we dramatically reduce the cost of modeling the parameter space with no sacrifice on either image diversity or fidelity. To illustrate this proposed plug-and-play framework, we construct variants of BasisGAN based on state-of-the-art conditional image generation networks, and train the networks by simply plugging in a basis generator, without additional auxiliary components, hyperparameters, or training objectives. The experimental success is complemented with theoretical results indicating how the perturbations introduced by the proposed sampling of basis elements can propagate to the appearance of generated images.

## 1 INTRODUCTION

Conditional image generation networks learn mappings from the condition domain to the image domain by training on massive samples from both domains. The mapping from a condition, e.g., a map, to an image, e.g., a satellite image, is essentially one-to-many as illustrated in Figure 1. In other words, there exists many plausible output images that satisfy a given input condition, which motivates us to explore multi-mode conditional image generation that produces diverse images conditioned on one single input condition.

One technique to improve image generation diversity is to feed the image generator with an additional latent code in the hope that such code can carry information that is not covered by the input condition, so that diverse output images are achieved by decoding the missing information conveyed through different latent codes. However, as illustrated in the seminal work Isola et al. (2017), encoding the diversity with an input latent code can lead to unsatisfactory performance for the following reasons. While training using objectives like GAN loss Goodfellow et al. (2014), regularizations like L1 loss Isola et al. (2017) and perceptual loss Wang et al. (2018) are imposed to improve both visual fidelity and correspondence to the input condition. However, no similar regularization is imposed to enforce the correspondence between outputs and latent codes, so that the network is prone to ignore input latent codes in training, and produce identical images from an input condition even with different latent codes. Several methods are proposed to explicitly encourage the network to take into account input latent codes to encode diversity. For example, Mao et al. (2019) explicitly maximizes the ratio of the distance between generated images with respect to the corresponding latent codes; while Zhu et al. (2017b) applies an auxiliary network for decoding the latent codes from the generative images. Although the diversity of the generative images is significantly improved, these methods experience drawbacks. In Mao et al. (2019), at least two samples generated from the same condition are needed for calculating the regularization term, which multiplies the memory footprint while training each mini-batch. Auxiliary network structures and training objectives in Zhu et al. (2017b) unavoidably increase training difficulty and memory footprint. These previously proposed methods usually require considerable modifications to the underlying framework.

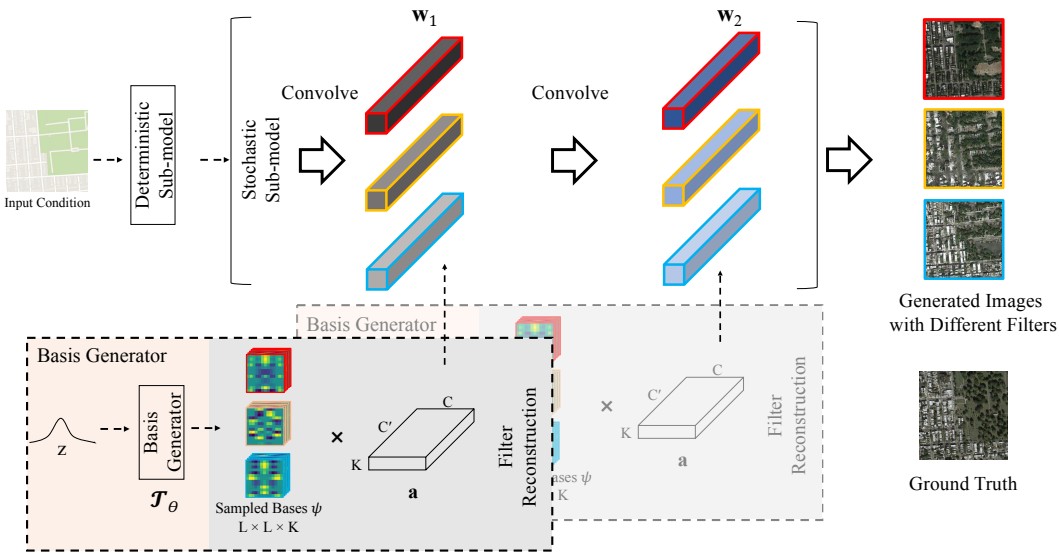

Figure 1: Illustration of the proposed BasisGAN. The diversity generated images are achieved by the parameter generation in the stochastic sub-model, where basis generators take samples from a prior distribution and generate low dimensional basis elements from the learned spaces. The sampled basis elements are linearly combined using the deterministic bases coefficients and used to reconstruct the convolutional filters. Filters in each stochastic layer are modeled with a separate basis generator. By convolving the same feature from the deterministic sub-model using different convolutional filters, images with diverse appearances are generated.

In this paper, we propose a stochastic model, *BasisGAN*, that directly maps an input condition to diverse output images, aiming at building networks that model the multi-mode intrinsically. The proposed method exploits a known observation that a well-trained deep network can converge to significantly different sets of parameters across multiple trainings, due to factors such as different parameter initializations and different choices of mini-batches. Therefore, instead of treating a conditional image generation network as a deterministic function with fixed parameters, we propose modeling the filter in each convolutional layer as a sample from filter space, and learning the corresponding filter space using a tiny network for efficient and diverse filter sampling. In Ghosh et al. (2018), parameter non-uniqueness is used for multi-mode image generation by training several generators with different parameters simultaneously as a multi-agent solution. However, the maximum modes of Ghosh et al. (2018) are restricted by the number of agents, and the replication increases memory as well as computational cost. Based on the above parameters non-uniqueness property, we introduce into a deep network stochastic convolutional layers, where filters are sampled from learned filter spaces. Specifically, we learn the mapping from a simple prior to the filter space using neural networks, here referred to as *filter generators*. To empower a deterministic network with multi-mode image generation, we divide the network into a deterministic sub-model and a stochastic sub-model as shown in Figure 1, where standard convolutional layers and stochastic convolutional layers with filter generators are deployed, respectively. By optimizing an adversarial loss, filter generators can be jointly trained with a conditional image generation network. In each forward pass, filters at stochastic layers are sampled by filter generators. Highly diverse images conditioned on the same input are achieved by jointly sampling of filters in multiple stochastic convolutional layers.

However, filters of a convolutional layer are usually high-dimensional while being together written as one vector, which makes the modeling and sampling of a filter space highly costly in practice in terms of training time, sampling time, and filter generator memory footprint. Based on the low-rank property observed from sampled filters, we decompose each filter as a linear combination of a small set of basis elements Qiu et al. (2018), and propose to only sample low-dimensional spatial basis elements instead of filters. By replacing filter generators with *basis generators*, the proposed method becomes highly efficient and practical. Theoretical arguments are provided on how perturbations introduced by sampling basis elements can propagate to the appearance of generated images.

The proposed BasisGAN introduces a generalizable concept to promote diverse modes in the conditional image generation. As basis generators act as plug-and-play modules, variants of BasisGAN can be easily constructed by replacing in various state-of-the-art conditional image generation net-

works the standard convolutional layers by stochastic layers with basis generators. Then, we directly train them without additional auxiliary components, hyperparameters, or training objectives on top of the underlying models. Experimental results consistently show that the proposed BasisGAN is a simple yet effective solution to multi-mode conditional image generation. We further empirically show that the inherent stochasticity introduced by our method allows training without paired samples, and the one-to-many image-to-image translation is achieved using a stochastic auto-encoder where stochasticity prevents the network from learning a trivial identity mapping.

Our contributions are summarized as follows:

- We propose a plug-and-played basis generator to stochastically generate basis elements, with just a few hundred of parameters, to fully embed stochasticity into network filters.
- Theoretic arguments are provided to support the simplification of replacing stochastic filter generation with basis generation.
- Both the generation fidelity and diversity of the proposed BasisGAN with basis generators are validated extensively, and state-of-the-art performances are consistently observed.

## 2 RELATED WORK

**Conditional image generation.** Parametric modeling of the natural image distribution has been studied for years, from restricted Boltzmann machines Smolensky (1986) to variational autoencoders Kingma & Welling (2013); in particular variants with conditions Oord et al. (2016); Sohn et al. (2015); Van den Oord et al. (2016) show promising results. With the great power of GANs Goodfellow et al. (2014), conditional generative adversarial networks (cGANs) Isola et al. (2017); Pathak et al. (2016); Sangkloy et al. (2017); Wang et al. (2018); Xian et al. (2018); Zhu et al. (2017a) achieve great progress on visually appealing images given conditions. However, the quality of images and the loyalty to input conditions come with sacrifice on image diversity as discussed in Zhu et al. (2017b), which is addressed by the proposed BasisGAN.

**Multi-mode conditional image generation.** To enable the cGANs with multi-mode image generation, pioneer works like infoGAN Chen et al. (2016) and pix2pix Isola et al. (2017) propose to encode the diversity in an input latent code. To enforce the networks to take into account input latent codes, Zhu et al. (2017b) deploys auxiliary networks and training objectives to impose the recovery of the input latent code from the generated images. MSGAN Mao et al. (2019) and DSGAN Yang et al. (2019) propose regularization terms for diversity that enforces a larger distance between generated images with respect to different input latent codes given one input condition. These methods require considerable modifications to the underlying original framework.

**Neural network parameters generating and uncertainty.** Extensive studies have been conducted for generating network parameters using another network since Hypernetworks Ha et al. (2016). As a seminal work on network parameter modeling, Hypernetworks successfully reduce learnable parameters by relaxing weight-sharing across layers. Followup works like Bayesian Hypernetworks Krueger et al. (2017) further introduce uncertainty to the generated parameters. Variational inference based methods like Bayes by Backprop Blundell et al. (2015) solve the intractable posterior distribution of parameters by assuming a prior (usually Gaussian). However, the assumed prior unavoidably degrades the expressiveness of the learned distribution. The parameter prediction of neural network is intensively studied under the context of few shot learning Bertinetto et al. (2016); Qiao et al. (2018); Wang et al. (2019), which aims to customize a network to a new task adaptively and efficiently in a data-driven way. Apart from few shot learning, Denil et al. (2013) suggests parameter prediction as a way to study the redundancy in neural networks. While studying the representation power of random weights, Saxe et al. (2011) also suggests the uncertainty and non-uniqueness of network parameters. Another family of network with uncertainty is based on variational inference Blundell et al. (2015), where an assumption of the distribution on network weights is imposed for a tractable learning on the distribution of weights. Works on studying the relationship between local and global minima of deep networks Haeffele & Vidal (2015); Vidal et al. (2017) also suggest the non-uniqueness of optimal parameters of a deep network.

## 3 STOCHASTIC FILTER GENERATION

A conditional generative network (cGAN) Mirza & Osindero (2014) learns the mapping from input condition domain $\mathcal{A}$ to output image domain $\mathcal{B}$ using a deep neural network. The conditional image generation is essentially a one-to-many mapping as there could be multiple plausible instances $\mathbf{B} \in \mathcal{B}$ that map to a condition $\mathbf{A} \in \mathcal{A}$ Zhu et al. (2017b), corresponding to a distribution $p(\mathbf{B}|\mathbf{A})$. However, the naive mapping of the generator formulated by a neural network $G : \mathbf{A} \to \mathbf{B}$ is deterministic, and is incapable of covering the distribution $p(\mathbf{B}|\mathbf{A})$. We exploit the non-uniqueness of network parameters as discussed above, and introduce stochasticity into convolutional filters through plug-and-play *filter generators*. To achieve this, we divide a network into two sub-models:

- A deterministic sub-model with convolutional filters $\phi$ that remain fixed after training;
- A stochastic sub-model whose convolutional filters $\mathbf{w}$ are sampled from parameter spaces modeled by neural networks $T$, referred to as filter generators, parametrized by $\theta$ with inputs $z$ from a prior distribution, e.g., $\mathcal{N}(0, I)$ for all experiments in this paper.

Note that filters in each stochastic layer are modeled with a separate neural network, which is not explicitly shown in the formulation for notation brevity. With this formulation, the conditional image generation becomes $G_{\phi,\theta} : \mathbf{A} \to \mathbf{B}$, with stochasticity achieved by sampling filters $\mathbf{w} = T_\theta(z)$ for the stochastic sub-model in each forward pass. The conditional GAN loss Goodfellow et al. (2014); Mirza & Osindero (2014) then becomes

$$
\begin{aligned}
\min_G \max_D V(D, G) =& \mathbb{E}_{\mathbf{A}\sim p(\mathbf{A}), \mathbf{B}\sim p(\mathbf{B}|\mathbf{A})}[\log(D(\mathbf{A}, \mathbf{B}))]+ \\
& \mathbb{E}_{\mathbf{A}\sim p(\mathbf{A}), z\sim p(z)}[\log(1 - D(\mathbf{A}, G_{\phi,\theta}(\mathbf{A}; T_\theta(z))))],
\end{aligned}
\tag{1}
$$

where $D$ denotes a standard discriminator. Note that we represent the generator here as $G_{\phi,\theta}(A; T_\theta(z))$ to emphasize that the generator uses stochastic filters $\mathbf{w} = T_\theta(z)$.

Given a stochastic generative network parametrized by $\phi$ and $\theta$, and input condition $\mathbf{A}$, the generated images form a conditional probability $q_{\phi,\theta}(\mathbf{B}|\mathbf{A})$, so that (1) can be simplified as

$$
\begin{aligned}
\min_G \max_D V(D, G) =& \mathbb{E}_{\mathbf{A}\sim p(\mathbf{A}), \mathbf{B}\sim p(\mathbf{B}|\mathbf{A})} \log D(\mathbf{A}, \mathbf{B})+ \\
& \mathbb{E}_{\mathbf{A}\sim p(\mathbf{A}), \mathbf{B}\sim q_{\phi,\theta}(\mathbf{B}|\mathbf{A})} \log[1 - D(\mathbf{A}, \mathbf{B})].
\end{aligned}
\tag{2}
$$

When the optimal discriminator is achieved, (2) can be reformulated as

$$
C(G) = \max_D V(D, G) = \mathbb{E}_{\mathbf{A}\sim p(\mathbf{A})}[-\log(4) + 2 \cdot JSD(p(\mathbf{B}|\mathbf{A})||q_{\phi,\theta}(\mathbf{B}|\mathbf{A}))],
\tag{3}
$$

where $JSD$ is the Jensen-Shannon divergence (the proof is provided in the supplementary material). The global minimum of (3) is achieved when given every sampled condition $\mathbf{A}$, the generator perfectly replicates the true distribution $p(\mathbf{B}|\mathbf{A})$, which indicates that by directly optimizing the loss in (1), conditional image generation with diversity is achieved with the proposed stochasticity in the convolutional filters.

To optimize (1), we train $D$ as in Goodfellow et al. (2014) to maximize the probability of assigning the correct label to both training examples and samples from $G_{\phi,\theta}$. Simultaneously, we train $G_{\phi,\theta}$ to minimize the following loss, where filter generators $T_\theta$ are jointly optimized to bring stochasticity:

$$
\mathcal{L} = \mathbb{E}_{\mathbf{A}\sim p(\mathbf{A},\mathbf{B}), z\sim p(z)}[\log(1 - D(\mathbf{A}, G_{\phi,\theta}(\mathbf{A}; T_\theta(z))))].
\tag{4}
$$

We describe in detail the optimization of the generator parameters $\{\phi, \theta\}$ in supplementary material Algorithm 1.

**Discussions on diversity modeling in cGANs.** The goal of cGAN is to model the conditional probability $p(\mathbf{B}|\mathbf{A})$. Previous cGAN models Mao et al. (2019); Mirza & Osindero (2014); Zhu et al. (2017b) typically incorporate randomness in the generator by setting $\mathbf{B} = G(\mathbf{A}, z)$, $z \sim p(z)$, where $G$ is a deep network with *deterministic* parametrization and the randomness is introduced via $z$, e.g., a latent code, as an extra *input*. This formulation implicitly makes the following two assumptions: (A1) The randomness of the generator is independent from that of $p(A)$; (A2) Each realization $\mathbf{B}(\omega)$ conditional on $\mathbf{A}$ can be modeled by a CNN, i.e., $\mathbf{B} = G^\omega(\mathbf{A})$, where $G^\omega$ is a draw from an ensemble of CNNs, $\omega$ being the random event. (A1) is reasonable as long as the source of variation to be modeled by cGAN is independent from that contained in $\mathbf{A}$, and the rational of

(A2) lies in the expressive power of CNNs for image to image translation. The previous model adopts a specific form of $G^\omega(A)$ via feeding random input $z(\omega)$ to $G$, yet one may observe that the most general formulation under (A1), (A2) would be to sample the generator itself from certain distribution $p(G)$, which is independent from $p(\mathbf{A})$. Since generative CNNs are parametrized by convolutional filters, this would be equivalent to set $\mathbf{B} = G(\mathbf{A}; \mathbf{w}), \mathbf{w} \sim p(\mathbf{w})$, where we use ";" in the parentheses to emphasize that what after is parametrization of the generator. The proposed cGAN model in the current paper indeed takes such an approach, where we model $p(\mathbf{w})$ by a separate filter generator network.

## 4    STOCHASTIC BASIS GENERATION

Using the method above, filters of each stochastic layer $\mathbf{w}$ are generated in the form of a high-dimensional vector of size $L \times L \times C' \times C$, where $L$, $C'$, and $C$ correspond to the kernel size, numbers of input and output channels, respectively. Although directly generating such high-dimensional vectors is feasible, it can be highly costly in terms of training time, sampling time, and memory footprint when the network scale grows. We present a throughout comparison in terms of generated quality and sample filter size in supplementary material Figure A.1, where it is clearly shown that filter generation is too costly to afford. In this section, we propose to replace filter generation with basis generation to achieve a quality/cost effect shown by the red dot in Figure A.1. Details on the memory, parameter number, and computational cost are also provided at the end of the supplementary material, Section G.

For convolutional filters, the weights $\mathbf{w}$ is a 3-way tensor involving a spatial index and two channel indices for input and output channel respectively. Tensor low-rank decomposition cannot be defined in a unique way. For convolutional filters, a natural solution then is to separate out the spatial index, which leads to depth-separable network architectures Chollet (2017). Among other studies of low-rank factorization of convolutional layers, Qiu et al. (2018) proposes to approximate a convolutional filter using a set of prefixed basis element linearly combined by learned reconstruction coefficients.

Given that the weights in convolutional layers may have a low-rank structure, we collect a large amount of generated filters and reshape the stack of $N$ sampled filters to a 2-dimensional matrix $\mathbf{F}$ with size of $J \times J'$, where $J = N \times L \times L$ and $J' = C' \times C$. We consistently observe that $\mathbf{F}$ is always of low effective rank, regardless the network scales we use to estimate the filter distribution. If we assume that a collection of generated filters observe such a low-rank structure, the following theorem proves that it suffices to generate bases in order to generate the desired distribution of filters.

**Theorem 1.** *Let $(\Omega, \mathbb{P})$ be probability space and $\mathbf{F} : \Omega \to \mathbb{R}^{L^2 \times C' \times C}$ a 3-way random tensor, where $\mathbf{F}$ maps each event $\omega$ to $\mathbf{F}^\omega(u, \lambda', \lambda)$,  $u \in [L] \times [L]$,  $\lambda' \in [C']$,  $\lambda \in [C]$. For each fixed $\omega$ and $u$, $\mathbf{F}^\omega(u) := \{\mathbf{F}^\omega(u, \lambda', \lambda)\}_{\lambda', \lambda} \in \mathcal{L}(\mathbb{R}^{C'}, \mathbb{R}^C)$. If there exists a set of deterministic linear transforms $a_k$, $k = 1, \cdots, K$ in $\mathcal{L}(\mathbb{R}^{C'}, \mathbb{R}^C)$ s.t. $\mathbf{F}^\omega(u) \in Span\{a_k\}_{k=1}^K$ for any $\omega$ and $u$, then there exists $K$ random vectors $\mathbf{b}_k : \Omega \to \mathbb{R}^{L^2}$, $k = 1, \cdots, K$, s.t. $\mathbf{F}(u, \lambda', \lambda) = \sum_{k=1}^K \mathbf{b}_k(u) a_k(\lambda', \lambda)$ in distribution. If $\mathbf{F}$ has a probability density, then so do $\{\mathbf{b}_k\}_{k=1}^K$.*

The proof of the theorem is provided in the supplementary material.

We simplify the expensive filter generation problem by decomposing each filter as a linear combination of a small set of basis elements, and then sampling basis elements instead of filters directly. In our method, we assume that the diverse modes of conditional image generations are essentially caused by the spatial perturbations, thus we propose to introduce stochasticity to the spatial basis elements. Specifically, we apply convolutional filer decomposition as in Qiu et al. (2018) to write $\mathbf{w} = \psi\mathbf{a}$, $\psi \in R^{L \times L \times K}$, where $\psi$ are basis elements, $\mathbf{a}$ are decomposition coefficients, and $K$ is a pre-defined small value, e.g., $K = 7$. We keep the decomposition coefficients $\mathbf{a}$ deterministic and learned directly from training samples. Instead of using predefined basis elements as in Qiu et al. (2018), we adopt a basis generator network $\mathcal{T}(\theta, z)$ parametrized by $\theta$, that learns the mapping from random latent vectors $z$ to basis elements $\psi$ with stochasticity. The basis generator networks are jointly trained with the main conditional image generation network in an end-to-end manner. Note that we inherit the term 'basis' from DCFNet Qiu et al. (2018) for the intuition behind the proposed framework, and we do not impose additional constrains such as orthogonality or linear independence to the generated elements. Sampling the basis elements $\psi$ using basis generators dramatically reduces the difficulty on modeling the corresponding probability distribution. The costly filter generators in Section 3 is now replaced by much more efficient basis generators, and stochastic filters are

then constructed by linearly combining sampled basis elements with the deterministic coefficients, $\mathbf{w} = \psi\mathbf{a} = \mathcal{T}(\theta, z)\mathbf{a}$. The illustration on the convolution filter reconstruction is shown as a part of Figure 1. As illustrated in this figure, BasisGAN is constructed by replacing convolutional layers with the proposed stochastic convolutional layers with basis generators, and the network parameters can be learned without additional auxiliary training objective or regularization.

## 5 EXPERIMENTS

In this section, we conduct experiments on multiple conditional generation task. Our preliminary objective is to show that thanks to the inherent stochasticity of the proposed BasisGAN, multi-mode conditional image generation can be learned without any additional regularizations that explicitly promote diversity. The effectiveness of the proposed BasisGAN is demonstrated by quantitative and qualitative results on multiple tasks and underlying models. We start with a stochastic auto-encoder example to demonstrate the inherent stochasticity brought by basis generator. Then we proceed to image to image translation tasks, and compare the proposed method with: regularization based methods DSGAN Yang et al. (2019) and MSGAN Mao et al. (2019) that adopt explicit regularization terms that encourages higher distance between output images with different latent code; the model based method MUNIT Huang et al. (2018) that explicitly decouples appearance with content and achieves diverse image generation by manipulating appearance code; and BicycleGAN Zhu et al. (2017b) that uses auxiliary networks to encourage the diversity of the generated images with respect to the input latent code. We further demonstrate that as an essential way to inject randomness to conditional image generation, our method is compatible with existing regularization based methods, which can be adopted together with our proposed method for further performance improvements. Finally, ablation studies on the size of basis generators and the effect of $K$ are provided in the supplementary material, Section E.

### 5.1 STOCHASTIC AUTO-ENCODER

The inherent stochasticity of the proposed BasisGAN allows learning conditional one-to-many mapping even without paired samples for training. We validate this by a variant of BasisGAN referred as stochastic auto-encoder, which is trained to do simple self-reconstructions with real-world images as inputs. Only L1 loss and GAN loss are imposed to promote fidelity and correspondence. However, thanks to the inherent stochasticity of BasisGAN, we observe that the network does not collapse to a trivial identity mapping, and diverse outputs with strong correspondence to the input images are generated with appealing fidelity. Some illustrative results are shown in Figure 2.

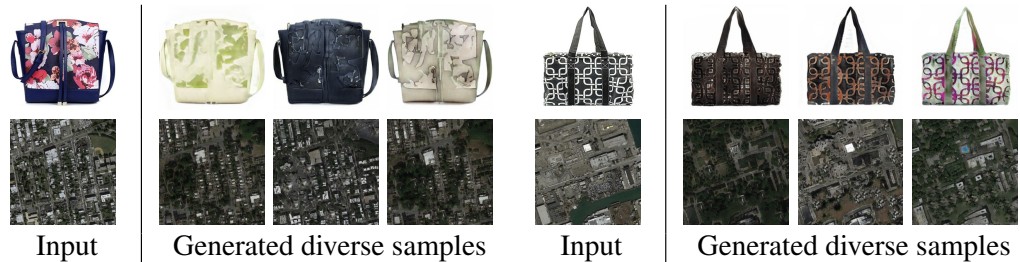

| Input | Generated diverse samples | Input | Generated diverse samples |

Figure 2: Stochastic auto-encoder: one-to-many conditional image generation without paired sample. The network is trained directly to reconstruct the input real-world images, and the inherent stochasticity of the proposed method successfully promotes diverse output appearances with strong fidelity and correspondence to the inputs.

### 5.2 IMAGE TO IMAGE TRANSLATION

To faithfully validate the fidelity and diversity of generated images, we follow Mao et al. (2019) to evaluate the performance quantitatively using the following metrics:

**LPIPS.** The diversity of generated images are measured using LPIPS Mao et al. (2019). LPIPS computes the distance of images in the feature space. Generated images with higher diversity give higher LPIPS scores, which are more favourable in conditional image generation.

**FID.** FID Heusel et al. (2017) is used to measure the fidelity of the generated images. It computes

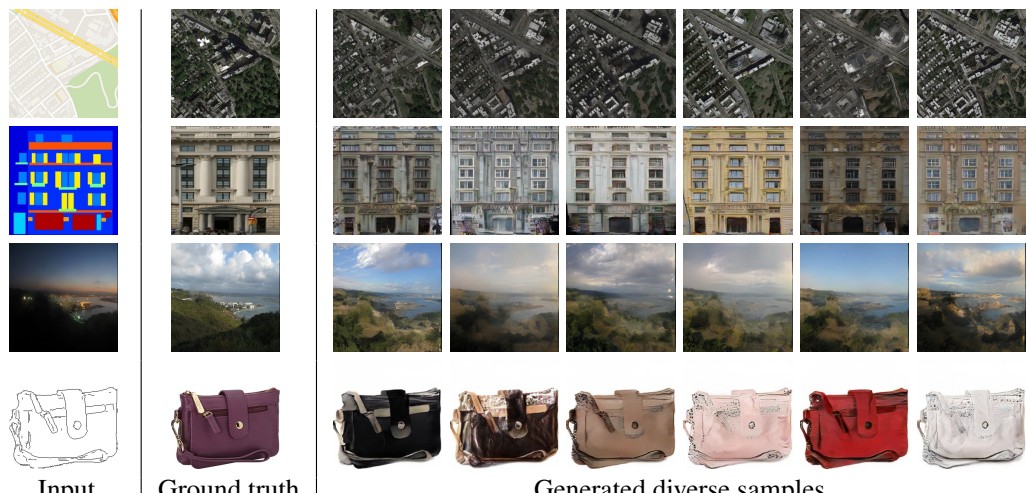

Input     Ground truth     Generated diverse samples

Figure 3: BasisGAN adapted from Pix2Pix. The network is trained without any auxiliary loss functions or regularizations. From top to bottom, the image to image translation tasks are: edges → handbags, edges → shoes, maps → satellite, nights → days, facades → buildings. Additional examples are provided in the supplementary material, Figure A.2.

the distance between the distribution of the generated images and the true images. Since the entire GAN family is to faithfully model true data distribution parametrically, lower FID is favourable in our case since it reflects a closer fit to the desired distribution.

**Pix2Pix → BasisGAN.** As one of the most prevalent conditional image generation network, Pix2Pix Isola et al. (2017) serves as a solid baseline for many multi-mode conditional image generation methods. It achieves conditional image generation by feeding the generator a conditional image, and training the generator to synthesize image with both GAN loss and L1 loss to the ground truth image. Typical applications for Pix2Pix include edge maps→shoes or handbags, maps→satellites, and so on. We adopt the ResNet based Pix2Pix model, and impose the proposed stochasticity in the successive residual blocks, where regular convolutional layers and convolutional layers with basis generators convolve alternatively with the feature maps. The network is re-trained from scratch directly without any extra loss functions or regularizations. Some samples are visualized in Figure 3. For a fair comparison with previous works Isola et al. (2017); Mao et al. (2019); Zhu et al. (2017b); Yang et al. (2019); Huang et al. (2018), we perform the quantitative evaluations on image to image translation tasks and the results are presented in Table 1. Qualitative comparisons are presented in Figure A.3. As discussed, all the state-of-the-art methods require considerable modifications to the underlying framework. By simply using the proposed stochastic basis generators as plug-and-play modules to the Pix2Pix model, the BasisGAN generates significantly more diverse images but still at comparable quality with other state-of-the-art methods. Moreover, as shown in Table A.3, BasisGAN reduces the number of trainable parameters comparing to the underlying methods thanks to the small number of basis elements and the tiny basis generator structures. While regularization based methods like Mao et al. (2019); Yang et al. (2019) maintain the parameter numbers of the underlying network models.

**Pix2PixHD → BasisGAN.** In this experiment, we report results on high-resolution scenarios, which particularly demand efficiency and have not been previously studied by other conditional image generation methods.

We conduct high resolution image synthesis on Pix2PixHD Wang et al. (2018), which is proposed to conditionally generate images with resolution up to $2048 \times 1024$. The importance of this experiment arises from the fact that existing methods Mao et al. (2019); Zhu et al. (2017b) require considerable modifications to the underlying networks, which in this case, are difficult to be scaled to very high resolution image synthesis due to the memory limit of modern hardware. Our method requires no auxiliary networks structures or special batch formulation, thus is easy to be scaled to large scale scenarios. Some generated samples are visualized in Figure 4. Quantitative results and comparisons

Table 1: Quantitative results on image to image translation. Diversity and fidelity are measured using LPIPS and FID, respectively. Pix2Pix Isola et al. (2017), BicycleGAN Zhu et al. (2017b), MSGAN Mao et al. (2019), and DSGAN Yang et al. (2019) are included in the comparisons. DSGAN adopts a different setting (denoted as 20s in the table) by generating 20 samples per input for computing the scores. We report results under both settings.

| Dataset | Labels → Facade | | | | | |
|---|---|---|---|---|---|---|
| Methods | Pix2Pix | BicycleGAN | MSGAN | BasisGAN | DSGAN (20s) | BasisGAN (20s) |
| Diversity ↑ | $0.0003 \pm 0.0000$ | $0.1413 \pm 0.0005$ | $0.1894 \pm 0.0011$ | $\mathbf{0.2648} \pm 0.004$ | 0.18 | $\mathbf{0.2594} \pm 0.004$ |
| Fidelity ↓ | $139.19 \pm 2.94$ | $98.85 \pm 1.21$ | $92.84 \pm 1.00$ | $\mathbf{88.7} \pm 1.28$ | 57.20 | $\mathbf{24.14} \pm 0.76$ |
| Datasets | Map → Satellite | | | | | |
| Methods | Pix2Pix | BicycleGAN | MSGAN | BasisGAN | DSGAN (20s) | BasisGAN (20s) |
| Diversity ↑ | $0.0016 \pm 0.0003$ | $0.1150 \pm 0.0007$ | $0.2189 \pm 0.0004$ | $\mathbf{0.2417} \pm 0.005$ | 0.13 | $\mathbf{0.2398} \pm 0.005$ |
| Fidelity ↓ | $168.99 \pm 2.58$ | $145.78 \pm 3.90$ | $152.43 \pm 2.52$ | $\mathbf{35.54} \pm 2.19$ | 49.92 | $\mathbf{28.92} \pm 1.88$ |

| Dataset | Edge → Handbag | | Edge → Shoe | |
|---|---|---|---|---|
| Methods | MUNIT | BasisGAN | MUNIT | BasisGAN |
| Diversity ↑ | $0.32 \pm 0.624$ | $\mathbf{0.35} \pm 0.810$ | $0.217 \pm 0.512$ | $\mathbf{0.242} \pm 0.743$ |
| Fidelity ↓ | $92.84 \pm 0.121$ | $\mathbf{88.76} \pm 0.513$ | $62.57 \pm 0.917$ | $64.17 \pm 1.14$ |

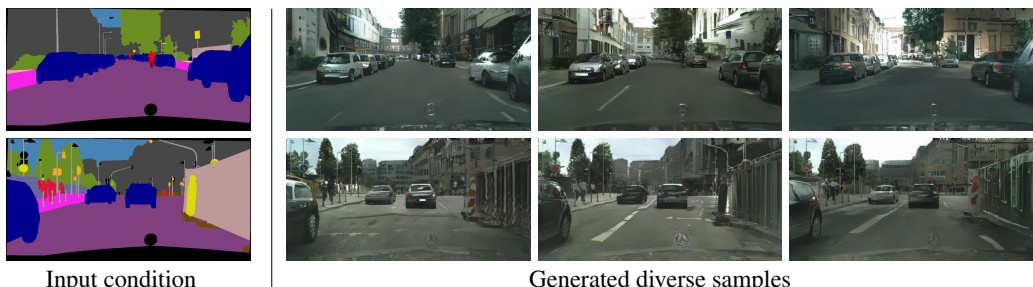

| Input condition | Generated diverse samples |
|---|---|

Figure 4: High resolution conditional image generation. Additional examples are provided in the supplementary material, Figure A.4.

against DSGAN Yang et al. (2019) are reported in Table 2. BasisGAN significantly improves both diversity and fidelity with little overheads in terms of training time, testing time, and memory.

**Image inpainting.** We conduct one-to-many image inpainting experiments on face images. Following Yang et al. (2019), centered face images in the celebA dataset are adopted and parts of the faces are discarded by removing the center pixels. We adopt the exact same network used in Yang et al. (2019) and replace the convolutional layers by layers with basis generators. To show the plug-and-play compatibility of the proposed BasisGAN, we conduct experiments by both training Basis-GAN alone and combining BasisGAN with regularization based methods DSGAN (BasisGAN + DSGAN). When combining BasisGAN with DSGAN, we feed all the basis generator in BasisGAN with the same latent code and use the distance between the latent codes and the distance between generated samples to compute the regularization term proposed in Yang et al. (2019). Quantitative results and qualitative results are in Table 3 and Figure 5, respectively. BasisGAN delivers good balance between diversity and fidelity, while combining BasisGAN with regularization based DSGAN further improves the performance.

Table 2: Quantitative results on high resolution image to image translation. Diversity and fidelity are measured using LPIPS and FID, respectively.

| Methods | Pix2PixHD | DSGAN | BasisGAN |
|---|---|---|---|
| Diversity ↑ | 0.0 | 0.12 | **0.168** |
| Fidelity ↓ | 48.85 | 28.8 | **25.12** |

Table 3: Quantitative results on face inpainting. Diversity and fidelity are measured using LPIPS and FID, respectively.

| Methods | DSGAN | BasisGAN | BasisGAN + DSGAN |
|---|---|---|---|
| Diversity ↑ | 0.05 | 0.062 | **0.073** |
| Fidelity ↓ | 13.94 | 12.88 | **12.82** |

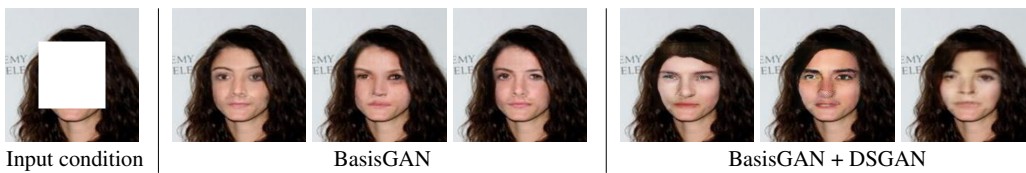

| Input condition | BasisGAN | BasisGAN + DSGAN |

Figure 5: Face inpainting examples.

## 6 CONCLUSION

In this paper, we proposed BasisGAN to model the multi-mode for conditional image generation in an intrinsic way. We formulated BasisGAN as a stochastic model to allow convolutional filters to be sampled from a filter space learned by a neural network instead of being deterministic. To significantly reduce the cost of sampling high-dimensional filters, we adopt parameter reduction using filter decomposition, and sample low-dimensional basis elements, as supported by the theoretical results here presented. Stochasticity is introduced by replacing deterministic convolution layers with stochastic layers with basis generators. BasisGAN with basis generators achieves high-fidelity and high-diversity, state-of-the-art conditional image generation, without any auxiliary training objectives or regularizations. Extensive experiments with multiple underlying models demonstrate the effectiveness and extensibility of the proposed method.

## 7 ACKNOWLEDGMENTS

Work partially supported by ONR, ARO, NGA, NSF, and gifts from Google, Microsoft, and Amazon.

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

# A  PROOF OF EQUATION (3)

*Proof.* Given (2) in Section 3, the minimax game of adversarial training is expressed as:

$$\min_G \max_D V(D, G) = \mathbb{E}_{\mathbf{A} \sim p(\mathbf{A}), \mathbf{B} \sim p(\mathbf{B}|\mathbf{A})} \log D(\mathbf{A}, \mathbf{B}) +$$

$$\mathbb{E}_{\mathbf{A} \sim p(\mathbf{A}), \mathbf{B} \sim q_{\phi,\theta}(\mathbf{B}|\mathbf{A})} \log[1 - D(\mathbf{A}, \mathbf{B})]$$

$$= \mathbb{E}_{\mathbf{A} \sim p(\mathbf{A})} \{ \mathbb{E}_{\mathbf{B} \sim p(\mathbf{B}|\mathbf{A})} \log D(\mathbf{A}, \mathbf{B}) + \mathbb{E}_{\mathbf{B} \sim q_{\phi,\theta}(\mathbf{B}|\mathbf{A})} \log[1 - D(\mathbf{A}, \mathbf{B})] \}. \tag{A.1}$$

By fixing $\mathbf{A}$ and only consider:

$$V' = \mathbb{E}_{\mathbf{B} \sim p(\mathbf{B}|\mathbf{A})} \log D(\mathbf{A}, \mathbf{B}) + \mathbb{E}_{\mathbf{B} \sim q_{\phi,\theta}(\mathbf{B}|\mathbf{A})} \log[1 - D(\mathbf{A}, \mathbf{B})]$$

$$= \int_{\mathbf{B}} p(\mathbf{B}|\mathbf{A}) \log D(\mathbf{A}, \mathbf{B}) + q_{\phi,\theta}(\mathbf{B}|\mathbf{A}) \log[1 - D(\mathbf{A}, \mathbf{B})] d\mathbf{B}. \tag{A.2}$$

The optimal discriminator $D^*$ in (A.2) is achieved when

$$D^*(\mathbf{A}, \mathbf{B}) = \frac{p(\mathbf{B}|\mathbf{A})}{p(\mathbf{B}|\mathbf{A}) + q_{\phi,\theta}(\mathbf{B}|\mathbf{A})}. \tag{A.3}$$

Given the optimal discriminator $D^*$, (A.2) is expressed as:

$$V' = \mathbb{E}_{\mathbf{B} \sim p(\mathbf{B}|\mathbf{A})} \log D^*(\mathbf{A}, \mathbf{B}) + \mathbb{E}_{\mathbf{B} \sim q_{\phi,\theta}(\mathbf{B}|\mathbf{A})} \log[1 - D^*(\mathbf{A}, \mathbf{B})]$$

$$= \mathbb{E}_{\mathbf{B} \sim p(\mathbf{B}|\mathbf{A})} [\log \frac{p(\mathbf{B}|\mathbf{A})}{p(\mathbf{B}|\mathbf{A}) + q_{\phi,\theta}(\mathbf{B}|\mathbf{A})}] + \mathbb{E}_{\mathbf{B} \sim q_{\phi,\theta}(\mathbf{B}|\mathbf{A})} [\log \frac{q_{\phi,\theta}(\mathbf{B}|\mathbf{A})}{p(\mathbf{B}|\mathbf{A}) + q_{\phi,\theta}(\mathbf{B}|\mathbf{A})}]$$

$$= -\log(4) + KL(p(\mathbf{B}|\mathbf{A})||\frac{p(\mathbf{B}|\mathbf{A}) + q_{\phi,\theta}(\mathbf{B}|\mathbf{A})}{2}) + KL(p(\mathbf{B}|\mathbf{A})||\frac{p(\mathbf{B}|\mathbf{A}) + q_{\phi,\theta}(\mathbf{B}|\mathbf{A})}{2})$$

$$= -\log(4) + 2 \cdot JSD(p(\mathbf{B}|\mathbf{A})||q_{\phi,\theta}(\mathbf{B}|\mathbf{A})) \tag{A.4}$$

where $KL$ is the Kullback-Leibler divergence. The minimum of $V'$ is achieved iff the Jensen-Shannon divergence is 0 and $p(\mathbf{B}|\mathbf{A}) = q_{\phi,\theta}(\mathbf{B}|\mathbf{A})$. And the global minimum of (A.1) is achieved when given every sampled $\mathbf{A}$, the generator perfectly replicate the conditional distribution $p(\mathbf{B}|\mathbf{A})$. $\square$

# B  PROOF OF THEOREM 4.1

*Proof.* We first consider the case when $\{a_k\}_{k=1}^K$ is a linearly independent set in the space of $\mathcal{L}(\mathbb{R}^{C'}, \mathbb{R}^C)$, which is finite dimensional (the space of $C'$-by-$C$ matrices). Then $\mathbf{F}^\omega(u)$ is in the span of $\{a_k\}_k$ for any $\omega, u$ means that there are unique coefficients $b(k; \omega, u)$ s.t.

$$\mathbf{F}^\omega(u) = \sum_{k=1}^K b(k; \omega, u) a_k,$$

and the vector $\{b(k; \omega, u)\}_k \in \mathbb{R}^k$ can be determined from $\mathbf{F}^\omega(u)$ by a (deterministic) linear transform. Since each entry $\mathbf{F}(u, \lambda', \lambda)$ is a random variable, i.e. measurable function on $(\Omega, \mathbb{P})$, then so is $b(k; \cdot, u)$ viewed as a mapping from $\Omega$ to $\mathbb{R}$, for each $k$ and $u$, due to that linear transform between finite dimensional spaces preserves measurability. For same reason, if $\mathbf{F}(u, \lambda', \lambda)$ has probability density, then so does each $b(k; \cdot, u)$. Letting $\{b(k; \cdot, u)\}_{u \in [L] \times [L]}$ be the random vectors $\mathbf{b}_k$ proves the statement.

When $\{a_k\}_{k=1}^K$ are linearly dependent, the dimensionality of the subspace where $\mathbf{F}^\omega(u)$ lie in is $\tilde{K} < K$. Suppose $\{\tilde{a}_k\}_{k=1}^{\tilde{K}}$ is a linearly independent set which spans the subspace, and $T : \mathbb{R}^{\tilde{K}} \to \mathbb{R}^K$ is the linear transform to map to $a_k$'s from $\tilde{a}_k$'s. Using the argument above, there exist random vectors $\tilde{b}_k$ s.t. $\mathbf{F} = \sum_{k=1}^{\tilde{K}} \tilde{b}_k \tilde{a}_k$, and using the pseudo-inverse of $T$ to construct random vectors $\{b_k\}_{k=1}^K$ we have that $\mathbf{F} = \sum_{k=1}^K b_k a_k$. This proves the existence of the $K$ random vectors $b_k$. $\square$

# C  PARAMETER OPTIMIZATION IN FILTER GENERATION

The optimization of the parameters $\{\phi, \theta\}$ in filter generation is presented in Algorithm 1.

---

**Algorithm 1** Optimization of the generator parameters $\{\phi, \theta\}$

---

**for** number of iterations **do**
- Sample a minibatch of $n$ pairs of samples $\{\mathbf{A}_1\mathbf{B}_1, \cdots, \mathbf{A}_n\mathbf{B}_n\}$.
- Sample $z \sim \mathcal{N}(0, I)$.
- Calculate the gradient w.r.t. the convolutional filters $\phi$ and $\mathbf{w}$ as in the standard setting

$$\Delta_\phi = \frac{\partial \mathcal{L}}{\partial \phi}, \Delta_\mathbf{w} = \frac{\partial \mathcal{L}}{\partial \mathbf{w}},$$

where $\mathcal{L} = \frac{1}{n}\sum_{i=1}^{n}[\log(1 - D(\mathbf{A}, G_{\phi,\theta}(\mathbf{A}; T_\theta(z))))]$.
- Calculate the gradient w.r.t. $\theta$ in the filter generator $\Delta_\theta = \Delta_\mathbf{w}\frac{\partial \mathbf{w}}{\partial \theta}$.
- Update the parameters $\phi$: $\phi \leftarrow \phi - \alpha\Delta_\phi$; $\theta$: $\theta \leftarrow \theta - \alpha\Delta_\theta$, where $\alpha$ is the learning rate.

**end for**

---

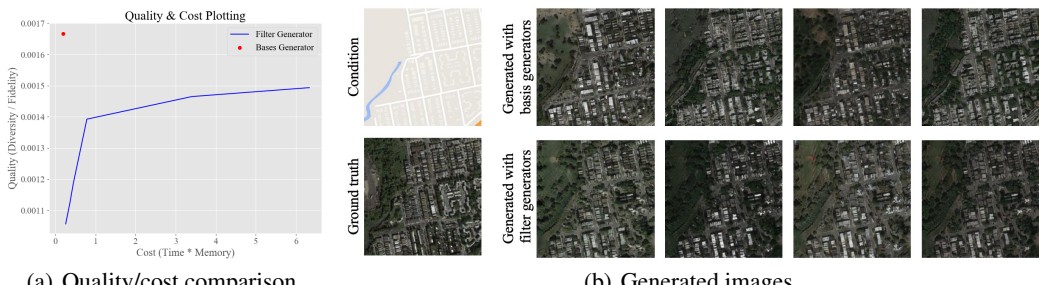

(a) Quality/cost comparison.          (b) Generated images.

Figure A.1: (a) shows the comparison between basis generation and filter generation in terms of quality and cost. In (b), top row shows images generated with basis generators (the red dot in (a)), bottom row shows images generated with filter generators at the highest cost (highest in (a)). Basis generation achieves better performance with significantly less cost comparing to filter generation. The quality metrics are introduced in Section 5.

## D    COMPUTATION COMPARISON

We present a throughout comparison in terms of generated quality and sample filter size in Figure A.1, where it is clearly shown that filter generation is too costly to afford, and basis generation achieves a significantly better quality/cost effect shown by the red dot in Figure A.1.

## E    ABLATION STUDIES

In this section, we perform ablation studies on the proposed BasisGAN, and evaluate multiple factors that can affect generation results. We perform ablation studies on BasisGAN adapted from the Pix2Pix model with the maps $\rightarrow$ satellite dataset.

**Size of basis generators.** We model a basis generator using a small neural network, which consists of several hidden layers and inputs a latent code sampled from a prior distribution. We consistently observe that a basis generator with a single hidden layer achieves the best performance while maintains fast basis generation speed. Here we perform further experiments on the size of intermediate layers and input latent code size, and the results are presented in Table A.1. It is observed that the size of a basis generator does not significantly effect the final performance, and we use the $64 + 64$ setting in all the experiments for a good balance between performances and costs.

**Number of basis elements K.** By empirically observing the low rank of generated filters, we use $K = 7$ in all the aforementioned experiments. We conduct further experiments to show the performances with larger $K$ and show the results in Table A.2. It is clearly shown that by increasing $K$, the quality of the generated images do not increase. And when $K$ gets larger, e.g, $K = 128$, even significantly degrades the diversity of the generated images.

Table A.1: Quantitative results with different sizes of input latent code and intermediate layer. $m+n$ denotes the size of latent code and intermediate layer.

| Dimensions | 16 + 16 | 32 + 32 | 64 + 64 | 128 + 128 | 256 + 256 | 512 + 512 |
|---|---|---|---|---|---|---|
| Diversity ↑ | 0.2242 | 0.2388 | 0.2417 | 0.2448 | 0.2452 | 0.2433 |
| Fidelity ↓ | 40.16 | 37.41 | 35.54 | 34.36 | 33.70 | 32.31 |

Table A.2: Quantitative results with different sizes of input latent code and intermediate layer. $m+n$ denotes the size of latent code and intermediate layer.

| K | 7 | 16 | 32 | 64 | 128 |
|---|---|---|---|---|---|
| Diversity ↑ | 0.2417 | 0.2409 | 0.2382 | 0.2288 | 0.2006 |
| Fidelity ↓ | 35.54 | 36.08 | 35.17 | 34.97 | 36.49 |

## F    QUALITATIVE RESULTS

### F.1    PIX2PIX →BASISGAN

Additional qualitative results for Pix2Pix → BasisGAN are presented in Figure A.2. Qualitative comparisons against MSGAN Mao et al. (2019) and DSGAN Yang et al. (2019) are presented in Figure A.3. We directly use the official implementation and the pretrained models provided by the authors. For each example, the first 5 generated samples are presented without any selection. For the satellite → map comparison, we often observe missing correspondence in the samples generated by DSGAN. BasisGAN consistently provides samples with diverse details and strong correspondence to the input conditions.

### F.2    PIX2PIXHD →BASISGAN

Additional qualitative results for Pix2PixHD →BasisGAN are presented in Figure A.4.

## G    SPEED, PARAMETER, AND MEMORY

We use PyTorch for the implementation of all the experiments. The training and testing are performed on a single NVIDIA 1080Ti graphic card with 11GB memory. The comparisons on testing speed, parameter number, and training memory are presented in Table A.3. The training memory is measured under standard setting with resolution of $256 \times 256$ for Pix2Pix, and $1024 \times 512$ for Pix2PixHD. Since we are using small number of basis elements (typically 7), and tiny basis generators, the overall trainable parameter number of the networks are reduced. Note that we only compute the parameter number of the generator networks since we do not adopt any change to the discriminators.

Table A.3: Speed in testing, memory usage in training, and overall trainable parameter numbers.

| Methods | Testing speed (s) | Training memory (MB) | Parameter number |
|---|---|---|---|
| Pix2Pix | 0.01017 | 1,465 | 11,330,243 |
| Pix2Pix → BasisGAN | 0.01025 | 1,439 | 10,261,763 |
| Pix2PixHD | 0.0299 | 8,145 | 182,546,755 |
| Pix2PixHD → BasisGAN | 0.0324 | 8,137 | 154,378,051 |

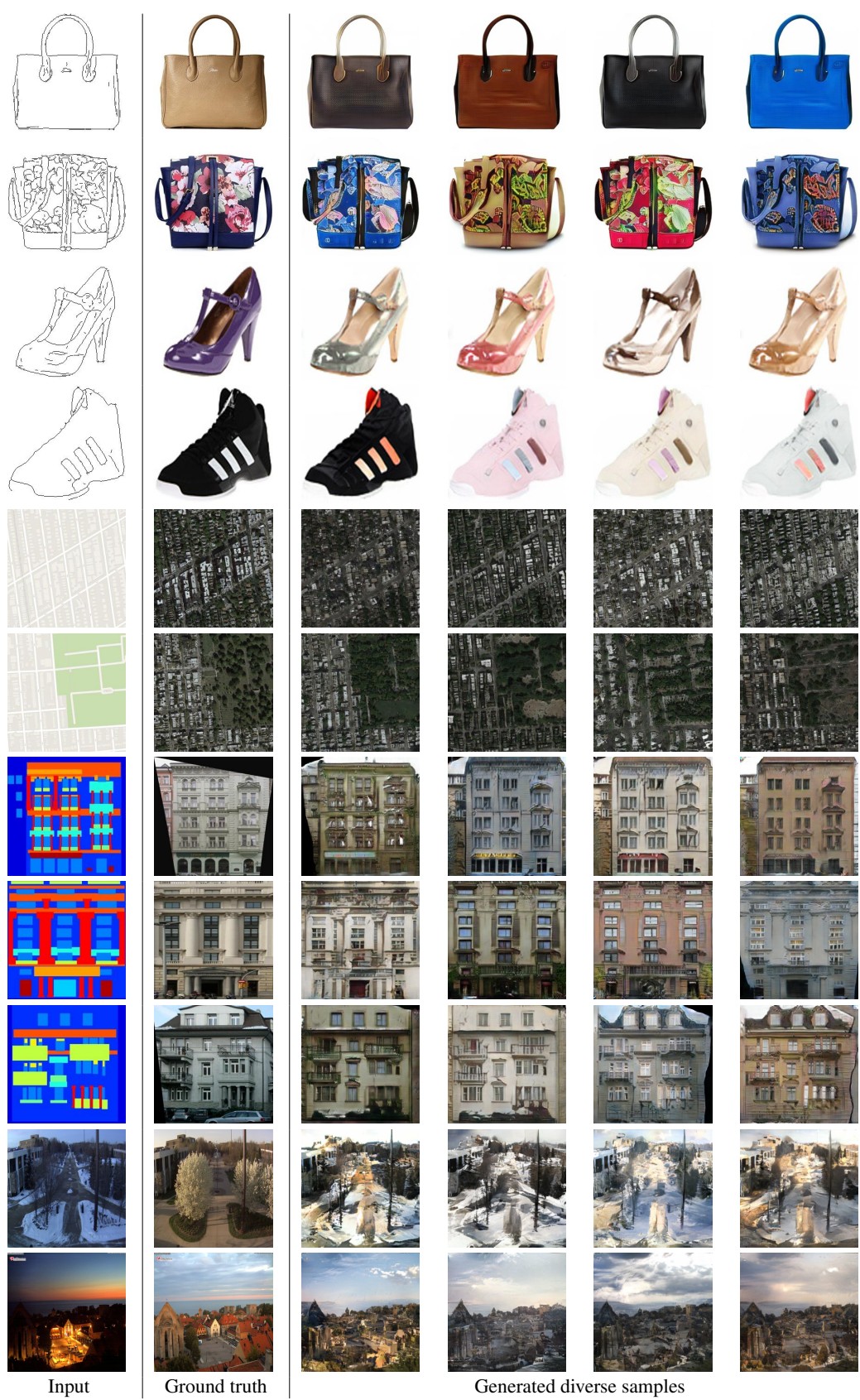

Input      Ground truth      Generated diverse samples

Figure A.2: Pix2Pix → BasisGAN.

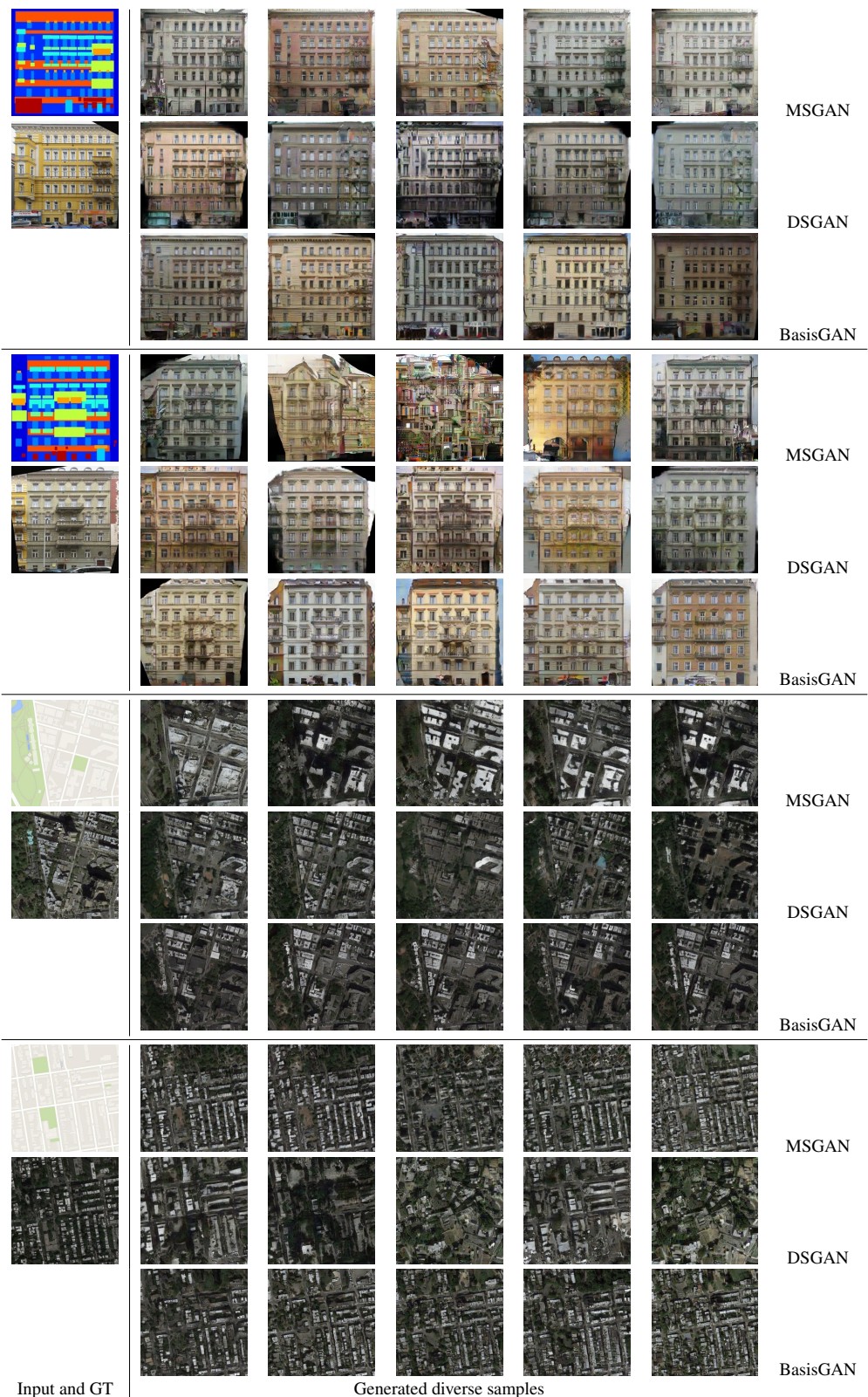

Input and GT     Generated diverse samples

Figure A.3: Qualitative comparisons with MSGAN and DSGAN. Please zoom in for details.

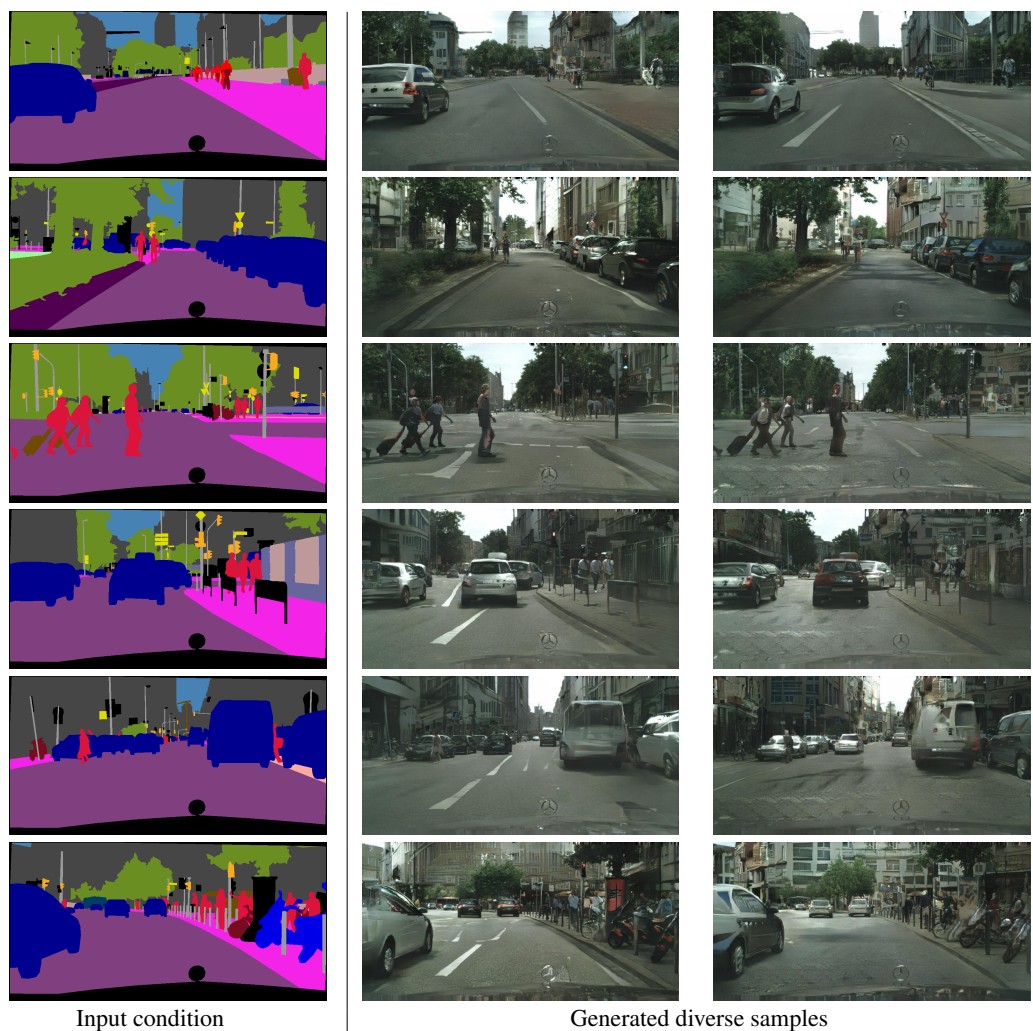

Input condition         Generated diverse samples

Figure A.4: Pix2PixHD → BasisGAN.

