# OpenReview forum: "Stochastic Conditional Generative Networks with Basis Decomposition"
_ICLR.cc/2020/Conference — Accept (Poster)_

### Official Review · AnonReviewer3 · 2019-10-23
**Official Blind Review #3**

**Rating:** 6

**Review:**

In this paper, the authors introduce BasisGAN, a novel method for introducing stochasticity in conditional GANs, i.e., a way of conducting one-to-many mappings. This is a good addition in the literature as: (a) most of the widely-used conditional GANs such as pix2pix (Isola et al., 2016) or pix2pixHD (Wang et al., 2018) are deterministic (i.e., for a specific input a single output is always generated), (b) it improves upon the current SOTA in one-to-many mappings, (c) it is very useful application-wise. As also stated in the paper, there is a number of applications where this method is handy (e.g., converting a sketch to images varying in colors, etc.).

I am leaning towards accepting this paper as this work is well-motivated and found the idea of using the basis generator to learn the bases for the generation of the parameters quite interesting. This is the main contribution and difference of this paper in comparison to DCFNet (Qiu et al., ICML 2018), where the bases are not learned.

Nevertheless, I have the following questions/requests:

- How can we tell that the generated bases are indeed bases (e.g., are they orthogonal?)
- Please report the number of parameters used in your implementation in comparison to the rest of the methods.
- Please provide qualitative results against the compared methods and especially against DSGAN (Qin et al., 2018).

**Experience Assessment:**

I have read many papers in this area.

**Review Assessment: Checking Correctness Of Derivations And Theory:**

I assessed the sensibility of the derivations and theory.

**Review Assessment: Checking Correctness Of Experiments:**

I carefully checked the experiments.

**Review Assessment: Thoroughness In Paper Reading:**

I read the paper at least twice and used my best judgement in assessing the paper.

---

> ### Author Response · Authors · 2019-11-15
> **Thanks and response to Reviewer #3**
>
>
> Thanks for the support and the valuable suggestions.
>
> *Basis generation
>
> In our training, we do not explicitly regularize the orthogonality of the generated basis elements. We provide additional discussion on linear independence in the response to R4, and have incorporated such additional details in Section 4 and Theorem 1.
>
>
> *Number of parameters
>
> The number of trainable parameters is updated in supplemental material Table A.3.  Adopting basis generator achieves fewer trainable parameters, thanks to the small number of basis elements and tiny structure of basis generators. This is critical to make the proposed framework and system trainable and manageable. Regularization based method like MSGAN and DSGAN do not reduce parameter numbers of the underlying network models. We have further discussed this in Section 5.2.
>
>
> *Qualitative results
>
> Thanks for your valuable suggestions and we have included qualitative comparisons in supplemental material Figure A.3.

---

### Official Review · AnonReviewer1 · 2019-11-01
**Official Blind Review #1**

**Rating:** 6

**Review:**

The paper proposes a new conditional GAN architecture. In particular, in order to allow for further diversity in conditional signal generation, the BasiGAN proposes to model the convolutional layers as a combination of basis which is stochastically sampled. The idea of the paper is interesting and some interesting experiments are presented. Nevertheless, I do not quite get why a set of predefined random basis would enforce more variability than the non-parametric way of training which is currently applied for conditional-GANs. If I get a convincing answer from the authors, I would definitely accept the paper (which otherwise is well-written and quite interesting to read).

**Experience Assessment:**

I have published in this field for several years.

**Review Assessment: Checking Correctness Of Derivations And Theory:**

I assessed the sensibility of the derivations and theory.

**Review Assessment: Checking Correctness Of Experiments:**

I assessed the sensibility of the experiments.

**Review Assessment: Thoroughness In Paper Reading:**

I read the paper at least twice and used my best judgement in assessing the paper.

---

> ### Author Response · Authors · 2019-11-15
> **Thanks and response to Reviewer #1**
>
> Thanks for the support.
>
> Instead of sampling predefined random basis, we learn in our network basis generators, in the form of tiny neural networks, that generate stochastic basis elements from random latent codes (details in Figure 1), which thus introduces great variability to the features and the output images without increasing dimensions to an unmanageable level.
>
> We have revised the last paragraph of Section 4 to further clarify this.

---

### Official Review · AnonReviewer4 · 2019-11-03
**Official Blind Review #4**

**Rating:** 6

**Review:**

The paper proposes a model for stochasticity for conditional image generation, building upon the previously available (DCFNet) results on composition of  convolutional filters out of the elements of the filter basis.

The idea of introducing stochasticity by convolutional filters into the conditional generative models seems to be novel and the reviewer thinks it could be of interest for the community.

The following remarks could be given to improve the presentation:
1) Theorem 1 is an existence theorem, so it does not give the procedure for construction of the basis. Does the construction procedure for the basis, described under the theorem formulation, meet the conditions of Theorem 1?
2) The Theorem 1 formulation states that “ If there exists a set of deterministic linear transforms”. Should the linear independence be stated as well as one of the theorem conditions ( so that the space dimensionality would indeed be K)?
3) The reviewer finds the structure of Section 4 confusing: it starts from the problem statement (first paragraph 'Using the method above, filters of each stochastic layer…’), then provides the description of the approach and only then outlines Theorem 1. It might be that stating Theorem 1 and then defining the method for generation of the basis (how exactly could we get to the basis? ) could improve readability of the paper. Essentially, the question is: is there any way to emphasise the procedure for filter generation and inform the reader in which circumstances these filters would be the basis (e.g. why it wouldn't be prone to the analogue of mode collapse when the filters do not effectively have enough diversity for linear independence)?
***
In addition to this list, it might be useful to provide some evidence on whether there is any inherent mechanism to regulate the diversity of filters and therefore of samples (so that to change the variability of the conditional samples from the model with the impact analogous to the one of temperature in Glow (Kingma et al, 2018)). If there is one, further experimental evidence, which shows the impact on diversity of filters, would contribute to improvement of the paper.


**Experience Assessment:**

I have read many papers in this area.

**Review Assessment: Checking Correctness Of Derivations And Theory:**

I assessed the sensibility of the derivations and theory.

**Review Assessment: Checking Correctness Of Experiments:**

I carefully checked the experiments.

**Review Assessment: Thoroughness In Paper Reading:**

I read the paper at least twice and used my best judgement in assessing the paper.

---

> ### Author Response · Authors · 2019-11-15
> **Thanks and response to Reviewer #4**
>
>
> Thanks for your support and all the insightful suggestions.
>
> *Construction procedure for the basis
>
> The network construction procedure meets the condition of Theorem 1, that is, we constrain the number of basis to a small number of K (e.g. K=7), which imposes low-rankness of the generated filters as stated in Theorem 1.
> As discussed in Section 4, when we directly generate random convolution filters, we consistently observe that the obtained filters are always of low effective rank. This observation of low-rankness motivates our construction of non-random (trainable) 1x1 convolutional layers ($a_k$'s) and randomly generated basis layers. In the revised version, we additionally present an ablation study in supplemental material Section E and Table A.2 to show that increasing K, i.e. using more basis elements, does not provide any performance improvement. We thus think K may serve as a parameter to regularize the model, also see below in "Model collapse".
>
>
> *Linear Independence
>
> Theoretically, linear independence is not needed for Theorem 1 to hold. In case that $a_k$'s are linearly dependent, it does not affect the existence result of the theorem, and the dimensionality of the subspace (the true rank) will be K' < K. We have revised the proof in supplemental material Section B to clarify this point. In practice, the trained $a_k$'s are of rank-K, due to that the choice of relatively small K is a tight restriction of the model.
>
>
> *Mode collapse
>
> In all experiments on real-world datasets, our model does not develop mode collapse as shown by the good diversity score. We are working to understand the precise relation between the fidelity-diversity trade-off. Based on the experiments (see supplemental material Table A.2) so far, we think the constraint rank K serves as a parameter to regularize the generative model. We will keep this as a direction of future efforts.
>
>
> The content of Section 4 is organized as following: we start with filter generation, which introduces stochasticity with considerable cost.
> Based on the low rank observation of generated filters, we further propose to decompose filters into bases with stochasticity and deterministic coefficients. And the decomposition is further supported by Theorem 1, which shows that it suffices to generate bases in order to generate the desired distribution of filters.
> We then provide detail on the process of basis generation, and how we construct the proposed BasisGAN with basis generators.

---

> > ### Comment · AnonReviewer4 · 2019-11-15
> > **Additional tests and explanations improve the paper**
> >
> > Many thanks to the authors for making the changes into the paper.
> >
> > " In the revised version, we additionally present an ablation study in supplemental material Section E and Table A.2 to show that increasing K, i.e. using more basis elements, does not provide any performance improvement."
> > The new ablation study improves the experimental assessment; however, as the best value is also the smallest, the question would appear why not to  decrease the basis size even further (would the performance still be the same or similar?)
> >
> > "Theoretically, linear independence is not needed for Theorem 1 to hold."
> > The reviewer  can see the point that it is not needed for the theorem but the name might be conflicting with the commonly used linear algebra definition for the basis, e.g.:
> > "A basis for a vector space is a sequence of vectors that is linearly independent and that spans the space." Jim Hefferon, 2017 Linear Algebra (p. 114)
> >
> > "In all experiments on real-world datasets, our model does not develop mode collapse as shown by the good diversity score."
> > The reviewer thinks that it might be hard to see it on real world datasets whether or not the model develops mode collapse; furthermore, how would it be possible to factor out capturing many diverse modes of the distribution which still do not cover many areas of the probability space? While the reviewer acknowledges that it might be impossible to do it for this revision due to the amount of time and work, it might be possible to demonstrate it by the experiments on the multimodal toy examples (8 gaussians, for example, as per  (Grathwohl et al (2019) / FFJORD)) for different values of K.

---

> > > ### Author Response · Authors · 2019-11-15
> > > **Additional explanations**
> > >
> > >
> > > Thanks for the additional valuable comments.
> > >
> > > * Smaller K
> > >
> > > The current experiment setting of K=7 was selected as performing the best after evaluating different K values starting from K=1 when preparing for the submission.
> > > Typically, when using K=5 or K=6, the performance drops slightly with good diversity but lower fidelity. When K drops to 4, the fidelity becomes much worse with obvious artifacts being observed in the generated images. And the network starts to have convergence problem in the adversarial training when K is small than 4.
> > >
> > >
> > > * Basis clarifications
> > >
> > > Thanks for pointing this out.
> > > Strictly linear dependence is allowed within the K elements, and in Theorem 1 we do not use the term ‘basis’. We inherit the term ‘basis’ from DCFNet more to illustrate the intuition behind the proposed framework. We added a further clarification in Section 4 to prevent confusions.
> > >
> > >
> > > * Mode collapse
> > >
> > > Thanks for the suggestions. The prevention of model collapse is actually not only reflected by the high diversity score, but also low FID score. As introduced in [1], FID is explicitly developed to compare the distance between the probability of observing real world data and the probability of generating model data. Low FID (especially for the map to satellite experiment) indicates the good coverage of generated samples to probability space of the target real-world data. Otherwise, if large areas of the probability space are missing, we cannot get such good scores on FID.
> > >
> > > We develop BasisGAN specifically for conditional image generation, where the diversity of the sample mainly comes from the spatial appearance of images. The setting is different from toy example as in [2]. We will definitely keep investigating the problem of mode collapse as a direction of future efforts.
> > >
> > >
> > > [1] GANs Trained by a Two Time-scale Update Rule Converge to a Local Nash Equilibrium, NIPS 2017.
> > > [2] FFJORD: Free-Form Continuous Dynamics for Scalable Reversible Generative Models, ICLR 2019.

---

### Decision · Program_Chairs · 2019-12-19

**Decision:**

Accept (Poster)

**Comment:**

Main content: BasiGAN, a novel method for  introducing stochasticity in conditional GANs
Summary of discussion:
reviewer1: interesting work and results on GANs. Reviewer had a question on pre-defned basis but i think it was answered by the authors.
reviewer3: interesting and novel work on GANS, wel-written paper and improves on SOTA. The main uestion is around bases again like reviewer 1, but it seems the authors have addressed this.
reviewer4: Novel interesting work. Main comments are around making Theorem 1 more theoretically correct, which it sounds like the authors addressed.
Recommendation: Poster. Well written and novel paper and authors addressed a lot of concerns.